# ACTION SHAPLEY: A TRAINING DATA SELECTION METRIC FOR HIGH PERFORMANCE AND COST EFFICIENT REINFORCEMENT LEARNING

## ABSTRACT

Numerous real-world reinforcement learning (RL) systems incorporate environment models to emulate the inherent dynamics of state-action-reward interactions. The efficacy and interpretability of such environment models are notably contingent upon the quality of the underlying training data. In this context, we introduce Action Shapley as an agnostic metric for the judicious and unbiased selection of training data. To facilitate the computation of Action Shapley, we present an algorithm specifically designed to mitigate the exponential complexity inherent in traditional Shapley value computations. Through empirical validation across four data-constrained real-world case studies, our proposed Action Shapley computation algorithm demonstrates a computational efficiency improvement exceeding 80% in comparison to conventional brute-force exponential time computations. Furthermore, our Action Shapley-based training data selection policy consistently outperforms ad-hoc training data selection methodologies across all examined case studies.

## 1 INTRODUCTION

We introduce Action Shapley, a training data selection metric for reinforcement learning (RL) environment model training (Sutton & Barto, 2018). It is inspired by the Shapley value (Shapley, 1953b;a) introduced by L.S. Shapley as a payoff function for different players in a stochastic game. There are several Shapley based approaches in supervised learning for data valuation (Ghorbani & Zou, 2019) and interpretability Lundberg & Lee (2017a). Compared to supervised learning, reinforcement learning brings additional complexity due to its interactive learning and stability problems such as the deadly triad (Sutton & Barto, 2018). Action Shapley is designed specifically to handle the problems related to reinforcement learning especially in a data-constrained and partially observable setting. In this paper, we focus on cloud system design problems such as virtual machine (VM) right-sizing (Derdus et al., 2019), load balancing (Mishra et al., 2020), database tuning (Bangare et al., 2016), and Kubernetes management (Rzadca et al., 2020). RL has been applied in VM right-sizing (Yazdanov & Fetzer, 2013), load balancing (Xu et al., 2019), database tuning (Wang et al., 2021), and Kubernetes management (Khaleq & Ra, 2021). Despite its benefits, the adoption of RL in cloud system design is stymied by data related issues such as data sparsity (Kamthe & Deisenroth, 2018), noisy environment (Dulac-Arnold et al., 2019), partial observability (Futoma et al., 2020), and irregular data sampling (Yildiz et al., 2021). The aforementioned issues in data quality therefore validate the usefulness of Action Shapley. This paper includes four case studies, as follows, specifically designed to validate the efficacy of Action Shapley.

- *VM Right-Sizing*: how to adjust vCPU count and memory size for a VM in order to bring its p50 CPU usage below a pre-assigned threshold?

- *Load Balancing*: how to adjust CPU and memory worker counts for a VM in order to bring its p5 CPU usage below a pre-assigned threshold?

- *Database Tuning*: how to adjust vCPU count and memory size for a database in order to bring its p90 CPU usage below a pre-assigned threshold?

- *Kubernetes (K8s) Management*: how to maximize write rates on a distributed database running on a Kubernetes cluster while keeping the p99.9 latency below a pre-assigned threshold?

The primary focus of these case studies revolves around cloud systems. However, it is imperative to emphasize a deliberate avoidance of relying on idiosyncratic attributes exclusive to cloud systems for the validation of Action Shapley's applicability across diverse contexts. Furthermore, the training data utilized in case studies, such as virtual machine (VM) right-sizing and database tuning, manifests notable distinctions, as elucidated in the Appendix. This variance in training data enhances our ability to assert the generalizability of our study, particularly within the realm of data-constrained and partially observable reinforcement learning (RL) systems. Conversely, the distinct advantage conferred by Action Shapley diminishes in RL systems characterized by high-fidelity, pre-existing simulation environments. Exemplary instances of such domains encompass video games such as Atari, AlphaGo, and StarCraft, as well as purpose-built simulated environments like MuJoCo for robotics, ToyText for puzzles, Box2D for locomotives, ns3-gym for networking, RecoGym for recommendation systems, AirSim for autonomous vehicles, among others.

## 2 METHODOLOGY

### 2.1 BACKGROUND AND NOTATION

In the realm of Reinforcement Learning (RL), the primary objective of a goal-seeking agent lies in the maximization of a cumulative reward score through dynamic interactions with its environment. These interactions are succinctly encapsulated in a temporal sequence denoted as $\langle state, action, reward \rangle$ trajectories, expressed as $\langle s_0, a_0, r_1, s_1, a_1, r_2, s_2, a_2, r_3... \rangle$. Under the assumption of a Markov decision process, the conditional joint probability distribution of reward ($r_t$) and state ($s_t$) is contingent solely upon the antecedent state and action, denoted as $p(s_t, r_t | s_{t-1}, a_{t-1})$. In the context of model-based RL systems, the underlying supervised model strives to capture this distribution. This necessitates a training corpus comprising two pairs: firstly, $(s_{t-1}, a_{t-1})$, followed by $(s_t, r_t)$. In the majority of cases, the reward score can be derived from the state: $s_t \mapsto r_t$. Consequently, a training data point is aptly represented as: $\mathcal{D} = \{(s_t^n; (s_{t-1}^n, a_{t-1}^n))\}_n$.

The concept of Action Shapley emerges as a mechanism designed to gauge the differential contribution of each data point within $\mathcal{D}$ to the resultant RL agent. This differential contribution is quantified through a valuation function, denoted as $\mathcal{U}$. Without loss of generality, $\mathcal{U}$ is defined as the cumulative reward garnered by the RL agent, thereby constituting an implicit function of the RL algorithm executed by the agent. In a general formulation, Action Shapley is expressed as $\phi(\mathcal{D}; \mathcal{U})$. To ensure equitable and robust valuation, we posit that $\phi$ adheres to the properties of nullity, symmetry, and linearity. Under these assumptions, the Action Shapley value pertaining to a specific training data point $\{k\}$ is delineated by Equation 1, as per established literature (Ghorbani & Zou, 2019; Shapley, 1953a; Tachikawa et al., 2018; Shapley, 1953b).

$$\phi_k = C_f \sum_{d \subseteq \mathcal{D} \setminus \{k\}} \frac{\mathcal{U}(d \cup \{k\}) - \mathcal{U}(d)}{\binom{n-1}{|d|}} \tag{1}$$

where, $C_f$ is an arbitrary constant. [1] While Equation 1's numerator gauges the distinct contribution of training data point $k$, it's denominator acts as the weight for $|d|$ (the number of elements in $|d|$). By summing over all possible dataset combinations, it provides a robust measure of the total differential contribution of $k$, surpassing the traditional Leave-One-Out (LOO) approach in marginal contribution and deduplication. In cases where the RL agent fails to achieve its goal for a training dataset, $d$, we assign $\mathcal{U}(d) \to$ null. Notably, the naive computation of Action Shapley (Equation 1) exhibits exponential time complexity ($\mathcal{O}(2^n)$) as it spans all subsets of $\mathcal{D}$ excluding $k$.

---

[1]Different symbols are explainedin the Notation section in the Appendix.

## 2.2 RANDOMIZED DYNAMIC ALGORITHM FOR ACTION SHAPLEY COMPUTATION

To expedite the computational efficiency of Action Shapley and circumvent the default exponential-time complexity, we introduce a randomized dynamic algorithm denoted as Algorithm 1. This algorithm demonstrates notably faster convergence on average. The approach capitalizes on the observation that a successful RL agent's environmental model necessitates a threshold number of training data points, denoted as the cut-off cardinality ($\theta$). Accordingly, Algorithm 1 employs a top-down accumulator pattern. Initialization involves considering the set encompassing all training data points, followed by a breadth-first traversal of the power set tree of the training data.

As depicted in Algorithm 1, the structure involves a nested $While$ loop syntax. The outer $While$ loop iterates over training data cardinalities in a descending order, while the inner $While$ loop iterates over all possible training data subsets for the specified cardinality. Within the inner loop, a check for RL agent failure is conducted (line 8 in Algorithm 1). In the event of a failure, the failure memoization accumulator is incremented by one (line 10 in Algorithm 1). The termination condition is met when the failure memoization accumulator, denoted as $mem$, reaches a user-defined threshold parameter, $\epsilon$. Upon termination, the cut-off cardinality (line 12) is set to be the cardinality where the failure condition (line 8) is first satisfied, incremented by one.

If the failure condition is not met, the marginal contribution of the training data ($k$) concerning the subset $d$ is added to the accumulation variable $sum$. This process of accumulating differential contributions continues until the termination condition is satisfied (line 17 in Algorithm 1).

Each training action is associated not only with its respective Action Shapley value but also possesses a unique *cut-off cardinality* value. The global *cut-off cardinality*, defined as the maximum among the *cut-off cardinality* values corresponding to distinct training data points, serves as an upper computational limit for Algorithm 1. The computational complexity of Algorithm 1 is situated between $\mathcal{O}(2^n)$, denoting the worst-case scenario, and $\mathcal{O}(\epsilon)$, reflecting the best-case scenario. The worst-case arises when the termination condition (line 11) is not met, leading to the exhaustive traversal of the entire combination tree. Conversely, the best-case occurs when the first $\epsilon$ evaluations fail, promptly reaching the termination condition (line 11). Significantly, the best-case scenario implies the indispensability of the training data point $k$, indicating that all agents fail to achieve the goal without the inclusion of data point $k$ in training. In such instances, data point $k$ is termed indispensable. It is noteworthy that a comparable approach involving the *cut-off cardinality* has been previously employed successfully in the machine learning literature (Tachikawa et al., 2018).

Combining these two extremes of $\mathcal{O}(2^n)$ and $\mathcal{O}(\epsilon)$, the performance of Algorithm 1 can be represented as a ratio of the exponential of global cut-off cardinality and the exponential of the total number of training data points subtracted from 1, as shown in Equation 2.

$$P_{comp} = 1 - \frac{2^{\theta_{k_{max}}}}{2^n} \qquad (2)$$

Action Shapley, by measuring the distinct contributions of various training data points, serves as a metric for selecting training data. A set exhibiting a higher average Action Shapley value outperforms a set with a lower average value. An ideal training dataset should comprise no more training data points than the specified global cut-off cardinality. A reduced number of training actions enhances computational efficiency. However, a minimum number of training data points is essential for constructing a successful reinforcement learning (RL) agent. Simply put, an inadequate number of training data points hinders the construction of an effective environment model, impeding goal achievement. The global cut-off cardinality represents the optimal balance, ensuring computational efficiency while meeting the minimum requirements for RL agent development. The proposed policy is underpinned by a theoretically sound foundation as an explanatory model for Shapley-based Additive Feature Attribution Methods Lundberg & Lee (2017a). We substantiate this hypothesis on training data selection and validate the effectiveness of Algorithm 1 with four distinct case studies.

## 3 DATA COLLECTION AND IMPLEMENTATION

We have implemented four distinct case studies to substantiate the effectiveness of Action Shapley, as previously delineated. In the investigations concerning VM right-sizing and load balancing, AWS

---

**Algorithm 1** Algorithm for Action Shapley Computation for a Training Action, $k$

---

**Input**: the total number of training data points: $n$; the set of all training data points: $\mathcal{D}$; RL algorithm used: $\mathcal{A}$; and valuation function for a training action subset without $k$: $\mathcal{U}(\mathcal{D}\backslash\{k\};\mathcal{A})$.
**Output**: Action Shapley value: $\phi_k$; cut-off cardinality: $\theta_k$ for action $k$.

**Parameter**: arbitrary constant: $C_f$; error bound: $\epsilon$.
**Variable**: training subset cardinality index: $i$; accumulator for marginal contributions for different subsets: $sum$; training subset: $d$; failure memoization: $mem$; termination indicator: $flag$

1: let $flag = 0$
2: let $i = n - 1$
3: let $sum = 0$
4: let $\theta_k = 1$
5: **while** iterate all cardinality greater than 1: $i > 1$ **do**
6:     let $mem = 0$
7:     **while** iterate all the sets of cardinality $i : d \in \mathcal{D}_i\backslash\{k\}$ **do**
8:         **if** $(\mathcal{U}(d \cup \{k\})$ is $null) \vee (\mathcal{U}(d)$ is $null)$ **then**
9:             $sum = sum$
10:             $mem = mem + 1$
11:             **if** $mem == \epsilon$ **then**
12:                 $\theta_k = i + 1$
13:                 $flag = 1$
14:                 **break** {get out of the inner loop (line 7)}
15:             **end if**
16:         **else**
17:             $sum = sum + C_f \frac{\mathcal{U}(d\cup\{k\})-\mathcal{U}(d)}{\binom{n-1}{|d|}}$
18:         **end if**
19:     **end while**
20:     **if** $flag == 1$ **then**
21:         **break** {get out of the outer loop (line 5)}
22:     **else**
23:         $i = i - 1$
24:     **end if**
25: **end while**
26: $\phi_k = sum$
27: **return** $\phi_k, \theta_k$

---

EC2 instances have been deployed, incorporating the *Stress* tool to facilitate dynamic workload simulations. In the context of the database tuning case study, we have employed a Nutanix AHV-based virtualized infrastructure, utilizing the *HammerDB* tool for dynamic SQL load simulation. The case study focusing on Kubernetes management entails the utilization of a Nutanix Kubernetes (K8s) engine cluster, with *cassandra-stress* employed for workload simulations.

In each case study, the training dataset is comprised of a collection of time series corresponding to different configurations, referred to as actions. These time series are measured over a 24-hour period at a 1-minute sampling rate. It is important to note that, across all case studies, the error bound ($\epsilon$) is consistently fixed at 1. Further elaboration on the specifics of data collection is available in Appendix.

Action Shapley is agnostic to the specific instantiations of the RL algorithm and the environment model algorithm. For the environment model, we use a radial basis function (RBF) network (He et al., 2019), along with principal component analysis (PCA)-based pre-training (Genovese et al., 2019), for the $(state_{prev}, action_{prev}) \rightarrow (state_{cur}, reward_{cur})$ mapping. Without loss of generality, we choose two different types of RL algorithms: SAC-PID (Yu et al., 2022) and PPO-PID (Shuprajhaa et al., 2022). The PID loop, as shown in Equation 3, is used as an RL action update policy based on the error term ($= (\text{threshold} - \text{aggregated state statistic})$), the time step ($(\delta t)$), and

Table 1: Action Shapley for VM right-sizing case study

| | AS SAC-PID | AS PPO-PID |
|---|---|---|
| $a_1$ | -3.96 | -3.90 |
| **$a_2$** | **-1.84** | **-1.81** |
| $a_3$ | ind. | ind. |
| $a_4$ | ind. | ind. |
| $a_5$ | ind. | ind. |

Table 2: Action Shapley for load balancing case study

| | AS SAC-PID | AS PPO-PID |
|---|---|---|
| $w_1$ | 0.57 | 0.57 |
| $w_2$ | 1.02 | 1.10 |
| **$w_3$** | **8.31** | **8.27** |
| $w_4$ | 3.61 | 3.61 |
| $w_5$ | 5.55 | 5.51 |

Table 3: Action Shapley for database tuning case study

| | AS SAC-PID | AS PPO-PID |
|---|---|---|
| $p_1$ | Ind. | ind. |
| $p_2$ | 1.11 | 1.12 |
| **$p_3$** | **3.42** | **3.37** |
| $p_4$ | 1.75 | 1.72 |
| $p_5$ | 0.19 | 0.21 |
| $p_6$ | -0.23 | -0.22 |

learning parameters, $[k_p, k_i, kd]$. The RL algorithm is designed to tune the parameters while maximizing the cumulative reward score.

$$a_i = a_{i-1} + [kp \quad ki \quad kd] \cdot \begin{bmatrix} e & (\delta t)e & \frac{e}{\delta t} \end{bmatrix} \tag{3}$$

For the VM right-sizing case study, we have five different data points: $\langle a_1 : (2, 2), a_2 : (2, 4), a_3 : (2, 8), a_4 : (4, 16), a_5 : (8, 32)\rangle$. Each pair represents (vCPU count, Memory Size). The training dataset consists of five time series of CPU usage, each with 1440 data points. While p50 is used for the aggregated state statistic, the goal of the RL agent is to bring the p50 CPU usage below 90%. The starting action for the RL loop is $(6, 14)$. The error bound, $\epsilon$, is set to 1.

For the load balancing case study, we have five different data points: $\langle w_1 : (8, 16), w_2 : (8, 12), w_3 : (8, 2), w_4 : (1, 2), w_5 : (1, 16)\rangle$. Each pair represents (# of CPU workers, # of memory workers). The training dataset consists of five time series of CPU usage, each with 1440 data points. While p5 is used for the aggregated state statistic, the goal of the RL agent is to bring the p5 CPU usage below 70%. The starting action for the RL loop is $(5, 10)$. The error bound, $\epsilon$, is set to 1.

For the database tuning case study, we have six different data points: $\langle p_1 : (1, 1), p_2 : (4, 4), p_3 : (6, 3), p_4 : (8, 4), p_5 : (8, 8), p_6 : (10, 10)\rangle$. Each pair represents (vCPU count, Memory Size). The training dataset consists of six time series of CPU usage, each with 1440 data points. While p90 is used for the aggregated state statistic, the goal of the RL agent is to bring the p90 CPU usage below 25%. The starting action for the RL loop is $(5, 2)$. The error bound, $\epsilon$, is set to 1.

In the Kubernetes Management case study, we employ 15 different data points $\langle r_1 : (1 \times 10^6, 10), r_2 : (1 \times 10^6, 25), r_3 : (1 \times 10^6, 50), r_4 : (1 \times 10^6, 75), r_5 : (1 \times 10^6, 100), r_6 : (2 \times 10^6, 10), r_7 : (2 \times 10^6, 25), r_8 : (2 \times 10^6, 50), r_9 : (2 \times 10^6, 75), r_{10} : (2 \times 10^6, 100), r_{11} : (3 \times 10^6, 10), r_{12} : (3 \times 10^6, 25), r_{13} : (3 \times 10^6, 50), r_{14} : (3 \times 10^6, 75), r_{15} : (3 \times 10^6, 100)\rangle$. Each training configuration is denoted by a pair representing the tuple (write rate, thread count). The training dataset encompasses 15 time series data sets, each comprising 1440 data points, measuring response latency. The aggregated state statistic is determined by the p99.9 metric, and the primary objective of the RL agent is to reduce the p99.9 latency to below 100ms. The RL loop commences with an initial action of $(2.9 \times 10^6, 95)$. The error bound, $\epsilon$, is set to 1.

In each of the aforementioned case studies, the primary objective for the RL agent is to attain a pre-assigned threshold for a state statistic. To achieve this, we formulate the reward function for the RL agent as: $(\text{threshold} - \text{aggregated state statistic}(t))$ if $\text{threshold} < \text{aggregated state statistic}(t)$. This reward function is designed to prompt the agent to efficiently meet the specified threshold.

## 4 RESULTS

### 4.1 ACTION SHAPLEY VALUES FOR TRAINING DATA

Table 1 displays the two Action Shapley (AS) values corresponding to two RL algorithms SAC-PID and PPO-PID, and five distinct training data points, denoted as $\langle a_1, a_2, a_3, a_4, a_5 \rangle$, within the VM right-sizing case study. Notably, it was identified that the training data points $a_3$, $a_4$, and

Table 5: This table summarizes the number of training data points and related parameters for four different case studies.

| Case Study | VM Right-Sizing | Load Balancing | Database Tuning | K8s Management |
|---|---|---|---|---|
| No of Training Data Points | 5 | 5 | 6 | 15 |
| Cut-off Cardinality | 4 | 3 | 4 | 5 |
| No of Indispensable Data Points | 3 | 0 | 1 | 0 |
| No of Data Points to be Chosen | 1 | 3 | 3 | 5 |
| No of Possible Data Points | 2 | 5 | 5 | 15 |

$a_5$ are indispensable. $a_2$ exhibits the highest Action Shapley values, specifically -1.84 and -1.81, respectively for SAC-PID and PPO-PID. The global cut-off cardinality, denoted as $\theta$, is set at 4, resulting in $P_{comp} = 50\%$ as defined by Equation 2. It is noteworthy that both SAC-PID and PPO-PID demonstrate congruent Action Shapley values and identical cut-off cardinality values.

Table 2 displays the two Action Shapley (AS) values corresponding to two RL algorithms SAC-PID and PPO-PID, and five distinct training data points, denoted as $\langle w_1, w_2, w_3, w_4, w_5 \rangle$, within the load balancing case study. $w_3$ exhibits the highest Action Shapley values, specifically 8.31 and 8.27, respectively for SAC-PID and PPO-PID. The global cut-off cardinality, denoted as $\theta$, is set at 3, resulting in $P_{comp} = 50\%$ as defined by Equation 2. It is noteworthy that both SAC-PID and PPO-PID demonstrate congruent Action Shapley values and identical cut-off cardinality values.

Table 3 displays the two Action Shapley (AS) values corresponding to two RL algorithms SAC-PID and PPO-PID, and six distinct training data points, denoted as $\langle p_1, p_2, p_3, p_4, p_5, p_6 \rangle$, within the database tuning case study. Notably, it was observed that the training data point $p_1$ is deemed indispensable, whereas the remaining five training data points are considered dispensable. $p_3$ exhibits the highest Action Shapley values, specifically 3.42 and 3.37, respectively for SAC-PID and PPO-PID. The global cut-off cardinality, denoted as $\theta$, is set at 4, resulting in $P_{comp} = 75\%$ as defined by Equation 2. It is noteworthy that both SAC-PID and PPO-PID demonstrate congruent Action Shapley values and identical cut-off cardinality values.

Table 4: Action Shapley for K8s management case study.

| | AS SAC-PID | AS PPO-PID |
|---|---|---|
| $r_1$ | -0.7 | 0.68 |
| $r_2$ | 0.53 | 0.54 |
| $r_3$ | 0.61 | 0.62 |
| $r_4$ | -0.13 | -0.12 |
| $r_5$ | 0.12 | 0.11 |
| $r_6$ | -0.7 | -0.7 |
| $r_7$ | -0.24 | -0.25 |
| $r_8$ | 0.65 | 0.65 |
| $r_9$ | 0.42 | 0.42 |
| $r_{10}$ | 0.08 | 0.07 |
| $r_{11}$ | -1.16 | -1.17 |
| $r_{12}$ | -0.25 | -0.24 |
| **$r_{13}$** | **0.77** | **0.77** |
| $r_{14}$ | -0.31 | -0.31 |
| $r_{15}$ | 0.019 | 0.019 |

Table 4 displays the two Action Shapley (AS) values corresponding to two RL algorithms SAC-PID and PPO-PID, and 15 distinct training data points, denoted as $\langle r_1, r_2, r_3, r_4, r_5, r_6, r_7, r_8, r_9, r_{10}, r_{11}, r_{12}, r_{13}, r_{14}, r_{15} \rangle$, within the K8s case study. Notably, $r_{13}$ exhibits the highest Action Shapley values, specifically 0.77 for both SAC-PID and PPO-PID. The global cut-off cardinality, denoted as $\theta$, is set at 5, resulting in $P_{comp} = 99.9\%$ as defined by Equation 2. It is noteworthy that both SAC-PID and PPO-PID demonstrate congruent Action Shapley values and identical cut-off cardinality values.

## 4.2 VALIDATION OF ACTION SHAPLEY BASED TRAINING DATA SELECTION POLICY

In the preceding sections of this paper, we introduced a training data selection policy, which can be succinctly summarized as follows: the optimal training dataset should include the maximum number of training data points, up to the predefined global cut-off cardinality, with the highest average Action Shapley value. The selection process is further refined when indispensable data points are considered. For instance, Table 5 provides a demonstration in the context of the VM right-sizing

case study. In this scenario, where we need to choose a single data point from 2 dispensable data points, there are $\binom{2}{1} = 2$ available options. In both the load balancing and database tuning cases, the task involves selecting 3 data points out of 5 dispensable data points, resulting in $\binom{5}{3} = 10$ potential choices for these specific case studies. Lastly, in the K8s management case study, where the selection involves 5 data points out of 15 dispensable data points, there are $\binom{15}{5} = 3,003$ possible choices.

Our empirical validation process involving four case studies consists of two steps. Firstly, we assess whether the best Action Shapley agent, generated from the training set with the highest average Action Shapley value and an element count matching the cut-off cardinality, achieves a higher cumulative reward compared to the worst Action Shapley agent. The worst Action Shapley agent is derived from the training set with the lowest average Action Shapley value and the same number of elements as the cut-off cardinality. Subsequently, we investigate whether the best Action Shapley agent consistently outperforms the majority of other agents. To answer this question, we

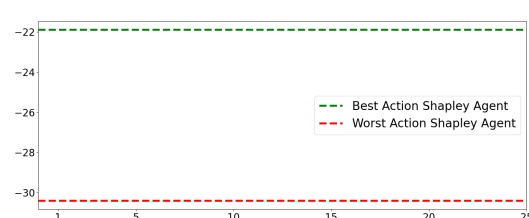

Figure 1: This plot compares the cumulative reward scores for the best Action Shapley agent vs the worst Action Shapley agent for the VM right-sizing case study.

conduct a series of 25 episodes, each involving multiple random selections. Each random selection is characterized by a training action set size equivalent to the cut-off cardinality. In light of the comparable Action Shapley values generated by both SAC-PID and SAC-PPO, we choose to utilize SAC-PID-based Action Shapley values in this section for the sake of convenience and without sacrificing generality.

For the VM right-sizing case study, our options for training action sets are limited to two. Consequently, our validation process is distilled into a comparative analysis between the best Action Shapley agent and the worst Action Shapley agent. Illustrated in Figure 1, the cumulative reward score for the superior Action Shapley agent registers at $-21.9$, while that for the inferior counterpart is recorded at $-30.4$. This discernible discrepancy in cumulative rewards substantiates the efficacy of the proposed training action selection policy.

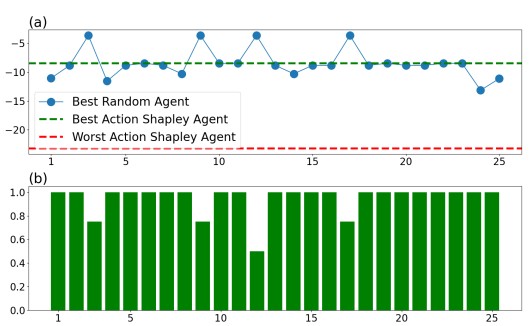

Figure 2: Validation of Action Shapley based selection policy for the load balancing case study. (a) Comparisons of cumulative rewards among the best Action Shapley agent, the worst Action Shapley agent, and the best of 4 random training action sets for 25 different episodes. (b) Fractions of agents based on 4 random training datasets with lower cumulative rewards than that of the best Action Shapley agent for 25 episodes.

In the context of the load balancing case study, a judicious selection of three training actions from a pool of five is imperative. The cumulative rewards, as depicted in Figure 2(a), elucidate a noteworthy performance disparity between the best and worst Action Shapley agents, with respective scores of $-8.49$ and $-23.2$. Notably, the optimal Action Shapley agent exhibits comparable efficacy to the highest achiever among randomly chosen sets across 25 episodes. Figure 2(b) further presents the proportion of randomly assembled training action sets resulting in agents with cumulative rewards surpassing that of the premier Action Shapley agent. The discernible trend reveals a mere 5 out of 100 randomly derived agents outperforming the optimal Action Shapley agent. Consequently, a confident assertion can be made that this case study effectively validates the viability of our chosen training action selection policy.

For the tuning case study, we faced the task of selecting four training datasets out of a pool of six. As depicted in Figure 3(a), the cumulative reward for the best Action Shapley agent, standing at $-2.42$, surpasses that of the worst Action Shapley agent, which is $-21$. Additionally, the figure illustrates that the performance of the best Action Shapley agent is comparable to the top-performing agent derived from four randomly selected training action sets across each of the 25 episodes. Figure 3(b) presents the fraction of randomly chosen training action sets that yield an agent with a cumulative reward lower than that of the best Action Shapley agent. While 31 out of 100 random selections perform better than the best Action Shapley agent, it's crucial to note that this difference is marginal, as evidenced by the top subplot comparing the best Action Shapley performance with the top performers in each of the 25 episodes. Consequently, we assert that this case study serves as a validation of our training action selection policy.

In our Kubernetes management case study, we had a pool of 3,003 options for selecting the training action set. Figure 4(a) displays the cumulative reward, indicating that the best Action Shapley agent achieved a score of $-499$, surpassing the worst Action Shapley agent, which scored $-621$. Furthermore, it demonstrates that the best Action Shapley agent's performance is on par with the top-performing agent from 30 randomly selected sets across each of the 25 episodes. Figure 4(b) presents the proportions of random training action sets resulting in an agent with a cumulative reward lower than that of the best Action Shapley agent. Notably, in this figure, it's observed that 125 out of 625 random selections outperform the best Action Shapley agent in terms of cumulative reward. However, it is crucial to highlight that despite this, the top performers from these random selections exhibit performance levels comparable to the best Action Shapley agent. Therefore, we confidently assert that this case study provides validation for our training action selection policy.

## 4.3 COMPARISON TO BASELINE RESULTS

All conducted case studies consistently validate the effectiveness of the proposed Action Shapley-based training data selection policy and computation algorithm. However, it is imperative to assess the efficacy of Action Shapley by benchmarking it against a baseline study that assumes the utilization of all available training data points for training the environment model. Specifically, in the VM right-sizing case study, the baseline study incorporates 5 training data points instead of the specified cutoff cardinality of 4. In the load balancing case study, the baseline study utilizes 5 training data points, deviating from the specified cutoff cardinality of 3. Similarly, in the database tuning case study, the baseline study integrates 6 training data points rather than the stipulated cutoff cardinality of 4. Lastly, in the K8s management case study, the baseline study employs 15 training data points, exceeding the defined cutoff cardinality of 5. In summary, the proposed Shapley analytics consistently lead to substantial reductions in the number of required training data points, ranging

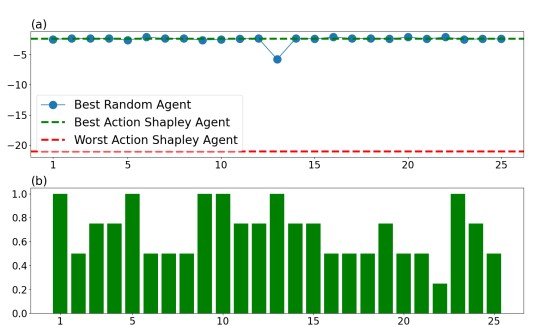

Figure 3: Validation of Action Shapley based selection policy for database tuning. (a) Comparisons of cumulative rewards among the best Action Shapley agent, the worst Action Shapley agent, and the best of 4 random training action sets for 25 different episodes. (b) Fractions of agents based on 4 random training datasets with lower cumulative rewards than that of the best Action Shapley agent for 25 episodes.

between 20% and 67%. The second evaluative dimension considers the cumulative reward attained by both the baseline agent and the optimal Action Shapley agent. Notably, in three of the four case studies (VM right-sizing, load balancing, and K8s management), the optimal Action Shapley agent demonstrates a significant performance advantage over the baseline agent. Specifically, for VM right-sizing, the values are -37.7 compared to -21.9; for load balancing, -9.1 compared to -8.49; and for K8s management, -561.4 compared to -499. In these instances, the optimal Action Shapley agent consistently outperforms the baseline agent by a substantial margin, ranging from 7% to 42%. However, in the database tuning case study, a marginal 3.9% superiority in cumulative reward is observed for the baseline agent.

## 5 RELATED WORK

The Shapley value has significantly influenced various fields, including economics (Roth, 1988), voting (Fatima et al., 2007), resource allocation (Du & Guo, 2016), and bargaining (Bernardi & Freixas, 2018). Within the realm of machine learning, methodologies inspired by the Shapley value have found application in data valuation (Ghorbani & Zou, 2019), model interpretability (Sundararajan & Najmi, 2020), and feature importance (Lundberg & Lee, 2017a). The Shapley value-based model explanation methodology falls under Additive Feature Attribution Methods (Lundberg & Lee, 2017b), which includes other methods such as LIME (Ribeiro et al., 2016), DeepLIFT (Shrikumar et al., 2016), and Layer-Wise Relevance Propagation (Bach et al., 2015).

In addressing the intricate challenge of training data selection in reinforcement learning (RL), compounded by the deadly triad (Sutton & Barto, 2018) of bootstrapping, functional approximation, and off-policy learning, researchers have explored diverse perspectives, including hindsight conditioning (Harutyunyan et al., 2019), return decomposition (Arjona-Medina et al., 2019), counterfactual multi-agent policy gradients (Foerster et al., 2018), corruption robustness (Zhang et al., 2022), optimal sample selection (Rachelson et al., 2011), active learning (Li et al., 2011), minimax PAC (Gheshlaghi Azar et al., 2013), $\epsilon$-optimal policy (Sidford et al., 2018), regret minimization (Jin et al., 2018), and statistical power (Colas et al., 2018).

The literature extensively explores the use of Shapley values for crediting agents in multi-agent RL (Li et al., 2021). Various methods for estimating Shapley value feature attributions are proposed in (Chen et al., 2023). Notably, there is a lack of dedicated studies on applying the Shapley value to select training data for reinforcement learning environment model training. Technical challenges in this regard include computational complexity, sample complexity, dynamic environmental conditions, data drift, feature interactions, interpretability limitations, and susceptibility to noise.

Action Shapley offers key advantages over alternative methods, including sample complexity, bias avoidance, interpretability, robustness, and resource efficiency (Lundberg & Lee, 2017a). Other data valuation techniques, like leave-one-out (LOO) testing (Cook & Weisberg, 1982), compare an agent's performance on the full dataset against its performance when trained on the dataset with one data point omitted. However, this method encounters difficulties with duplicated data in datasets.

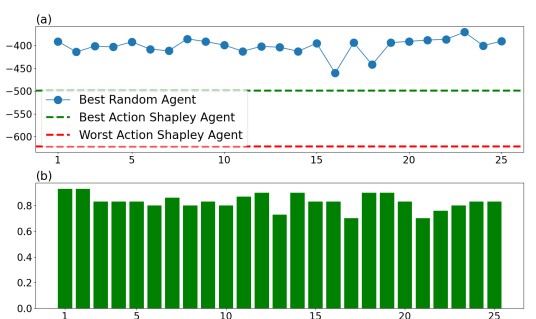

Figure 4: Validation of Action Shapley based selection policy for K8s management. (a) Comparisons of cumulative rewards among the best Action Shapley agent, the worst Action Shapley agent, and the best of 30 random training action sets for 25 different episodes. (b) Fractions of agents based on 30 random training datasets with lower cumulative rewards than that of the best Action Shapley agent for 25 episodes.

## 6 CONCLUSION

This paper introduces Action Shapley as a metric for selecting reinforcement learning training data. To address the inherent exponential time complexity, a randomized algorithm is proposed for computing Action Shapley. The effectiveness of this randomized algorithm and the associated training data selection policy is validated using four comprehensive case studies. The motivation behind this research stems from the critical role that life cycle management of reinforcement learning training data plays in distinguishing the performance of RL agents. We anticipate that the adoption of Action Shapley will facilitate real-world applications of reinforcement learning. Future endeavors will involve conducting additional real-world case studies, encompassing areas such as Kubernetes horizontal scaling, distributed machine learning training, database replication, and reinforcement learning-based fine-tuning of large language models (RLHF).

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

# A   APPENDIX

## A.1   NOTATION

| | |
|---|---|
| $s$ | Environment state |
| $a$ | Agent action |
| $r$ | Reward received by an agent |
| $\mathcal{D}$ | Training data set |
| $\mathcal{A}$ | RL algorithm |
| $\mathcal{U}$ | Evaluation function for resulting RL agent |
| $\phi_k$ | Action Shapley for a training data point $k$ |
| $C_f$ | Arbitrary constant for Action Shapley computation |
| $n$ | Number of data points in a data set $\mathcal{D}$ |
| $\epsilon$ | Error bound |
| $mem$ | Local variabel for failure memoization |
| $\theta_k$ | Cut-off cardinality |
| $\theta_{k_{max}}$ | Global cut-off cardinality |
| $k_p$ | Proportional parameter in PID |
| $k_i$ | Integral parameter in PID |
| $k_d$ | Differential parameter in PID |
| $\binom{n}{r}$ | n choose r |
| $|d|$ | Number of elements in a set, $\{d\}$ |
| $e$ | the difference between threshold and aggregated state statistic |
| $\delta t$ | the time step size in reinforce learning loop |

## A.2   DATA COLLECTION FOR THE VM RIGHT-SIZING CASE STUDY

VM right-sizing entails a decision problem that recognizes minimum possible VM size without compromising a pre-assigned performance threshold. In this case study, we use CPU utilization as the state variable. The training action is represented by a pair of (# of CPU cores, memory size (GB)). The five training actions used are: $\{a_1 : (2,2), a_2 : (2,4), a_3 : (2,8), a_4 : (4,16), a_5 : (8,32)\}$. Figure 5(a) shows training action samples and corresponding aggregated state statistics (i.e., 50-th percentile as mentioned in Table 6). The environment runs a rectangular workload, simulated using *stress* library. The simulated rectangular workload has a time period of 600 s: a high-stress, peak phase of 500 s is followed by a `sleep` phase of 100 s. The peak of the workload uses the stress command: $\{sudo\ stress\ –io\ 4\ –vm\ 2\ –vm\text{-}bytes\ 1024M\ –timeout\ 500s\}$. Essentially, the peak is running 4 *I/O* stressors and 2 VM workers spinning on malloc with 1024 MB per worker for 500 s. This workload runs for 24 hr. with a 1 minute sampling interval. Overall, five time series data for CPU utilization together represent the state space. The goal of the RL agent is to choose the most cost-efficient configuration to bring down the aggregated state statistic below a pre-assigned threshold of 90% without violating constraints. Until the goal is satisfied, the RL agent is penalized in every time step by the deviation of the aggregated state statistic from its set point. As shown in Table 6, two other parameters are: Initial Action $= (6, 14)$ and $\epsilon = 1$.

## A.3 Data Collection for the Load Balancing Case Study

Load balancing deals with a decision problem that recognizes how to distribute traffic/loads to different servers. For this, it is important to estimate the dynamic capacity of a VM. In this case study, we use CPU utilization as the state variable. The action space is represented by a pair of (# of CPU workers, # of memory workers). The five training actions used are: $\{w_1 : (8, 16), w_2 : (8, 12), w_3 : (8, 2), w_4 : (1, 2), w_5 : (1, 16)\}$. Figure 5(b) shows training action samples and corresponding aggregated state statistics (i.e., 5-th percentile as mentioned in Table 6). The environment for this case study is an AWS t3a.medium EC2 VMs with 2 vCPUs, 4GB RAM. Each training workload has a time period of 600 seconds: a high-stress, peak phase of 500 seconds is followed by an inactive `sleep` phase of 100 seconds. Here are five different training workloads during high-stress phases: {*stress –cpu 8 –io 4 –vm 16 –vm-bytes 1024M –timeout 500s*}, {*stress –cpu 8 –io 4 –vm 12 –vm-bytes 1024M –timeout 500s*},{*stress –cpu 8 –io 4 –vm 2 –vm-bytes 1024M –timeout 500s*}, {*stress –cpu 1 –io 4 –vm 2 –vm-bytes 1024M –timeout 500s*}, and {*stress –cpu 1 –io 4 –vm 16 –vm-bytes 1024M –timeout 500s*}. The only differences among these workloads are the number of cpu workers and the number of memory workers.Each of these workloads is run for 3 hours with 5-minute sampling interval. Overall, the state space consists of five time series for CPU utilization metrics. The goal of the RL agent is to choose the highest possible workload intensity by packing more CPU and memory workers without violating a pre-assigned performance goal of keeping the 5-th percentile of CPU utilization below 70%. Until the goal is satisfied, the RL agent is penalized in every time step by the deviation of the aggregated state statistic from its set point. As shown in Table 6, two other parameters are: Initial Action = $(5, 10)$ and $\epsilon = 1$.

## A.4 Data Collection for the Database Tuning Case Study

Two critical SQL database tuning parameters are the number of server CPU cores and server memory size (GB). The tuning of these parameters requires painstaking trial and error which can be replaced with RL for precision and efficiency. In this case study, we use CPU utilization as the state variable. The action space is represented by a pair of (# of CPU cores, memory size (GB)).The six training actions used are: $\{p_1 : (1, 1), p_2 : (4, 4), p_3 : (6, 3), p_4 : (8, 4), p_5 : (8, 8), p_6 : (10, 10)\}$. Figure 5(c) shows training action samples and corresponding aggregated state statistics (i.e., 90-th percentile as mentioned in Table 6). The goal of the RL agent is to bring the aggregated state statistics below $25\%$. Against each training action, a SQL workload is simulated using *HammerDB*. The corresponding server utilization data is collected for a duration of 172 minutes at a sampling interval of 30 seconds. The workload is triggered by a HammerDB-hosted Tcl code which simulates *two* SQL warehouse building processes by *two* SQL virtual workers. Overall, we have six time series data against each database server configurations as the state space. Until the goal is satisfied, the RL agent is penalized in every time step by the deviation of the aggregated state statistic from its set point. As shown in Table 6, two other parameters are: Initial Action = $(5, 2)$ and $\epsilon = 1$.

## A.5 Data Collection for the Kubernetes Management Case Study

Kubernetes management often deals with a decision problem that recognizes the optimal workload parameters for a given Kubernetes cluster to maintain certain service level agreement criteria. For the given case study, we use Cassandra database workload with the thread count and lines written as the parameters. The goal of the RL agent is to maintain the p99.9 latency below a pre-assigned threshold of 100 milliseconds. The initial action points are $2.9 \times 10^6$ for written line count and 95 for thread count. For training, we use 15 different pairs spanned by three written line counts: $\{1 \times 10^6, 2 \times 10^6, 3 \times 10^6\}$ and five thread counts of $\{10, 25, 50, 75, 100\}$. Figure 5(d) shows training action samples and corresponding aggregated state statistics (i.e., 99.9-th percentile as mentioned in Table 6). We set $\epsilon = 1$.

## A.6 Validation of Training Action Selection Policy

We have proposed an optimal training action selection policy: the best possible training action set includes as many training actions as the global cut-off cardinality with the highest possible Action Shapley values. On the other hand, the worst possible training action set includes as many training actions as the global cut-off cardinality with the lowest possible Action Shapley values. We validate this policy for four different case studies.

Table 6: Salient details of three different case studies

| Case Study | VM Right-Sizing | Load Balancing | Database Tuning | K8s Management |
|---|---|---|---|---|
| *State* | p50 CPU usage | p5 CPU usage | p90 CPU usage | p99.9 latency |
| *Threshold* | $\leq 90\%$ | $\leq 70\%$ | $\leq 25\%$ | $\leq 100ms$ |
| *Initial Action* | $(6, 14)$ | $(5, 10)$ | $(5, 2)$ | $(2.9 \times 10^6, 95)$ |
| $\epsilon$ | 1 | 1 | 1 | 1 |

### A.6.1 VALIDATION OF TRAINING ACTION SELECTION POLICY FOR THE VM RIGHT-SIZING CASE STUDY

Based on the Action Shapley computation, we determine that the cut-off cardinality is equal to 4 with 5 total number of training actions. It indicates $P_{comp} = 50\%$. With $a_3, a_4, a_5$ being the indispensable actions, $\langle a_2, a_3, a_4, a_5 \rangle$ has the highest possible mean Action Shapley value. On the other hand, $\langle a_1, a_3, a_4, a_5 \rangle$ has the lowest possible mean Action Shapley value. As we hypothesize, the former action set produced an RL agent with higher cumulative award of $-21.9$ vs $-37.75$. The former has a lower convergence time too: 13 vs 16. In comparison to the baseline RL agent (cumulative reward of $-30.4$ and convergence time of 21), both best and worst agents performs better.

### A.6.2 VALIDATION OF TRAINING ACTION SELECTION POLICY FOR THE LOAD BALANCING CASE STUDY

Based on the Action Shapley computation, we determine the cut-off cardinality is equal to 3 with 5 totral number of training actions, indicating that $P_{comp} = 75\%$. $\langle w_3, w_4, w_5 \rangle$ has the highest possible mean Action Shapley value. On the other hand, $\langle w_1, w_2, w_4 \rangle$ has the lowest possible mean Action Shapley value. As we hypothesize, the former action set produced an RL agent with a higher cumulative award of $-8.49$ vs $-23.2$. The former has a lower convergence time too: 10 vs 29. In comparison to the baseline RL agent (cumulative reward of $-9.1$ and convergence time of 12), only the best agent performs better.

### A.6.3 VALIDATION OF TRAINING ACTION SELECTION POLICY FOR THE DATABASE TUNING CASE STUDY

Based on the Action Shapley computation, we determine the cut-off cardinality is equal to 4 from 6 total actions. This indicates $P_{comp} = 75\%$. With $p_1$ being the indispensable actions, $\langle p_1, p_2, p_3, p_4 \rangle$ has the highest possible mean Action Shapley value. On the other hand, $\langle p_1, p_2, p_5, p_6 \rangle$ has the lowest possible mean Action Shapley value. As we hypothesize, the former action set produced an RL agent with a higher cumulative award of $-2.42$ vs $-21$. The former has a lower convergence time too: 7 vs 16. In comparison to the baseline RL agent (cumulative reward of $-2.33$ and convergence time of 6), both the best and worst agent agents performs poorly. In fact, the baseline agent performs marginally better than the best agent chosen by the Action Shapley policy.

### A.6.4 VALIDATION OF TRAINING ACTION SELECTION POLICY FOR THE KUBERNETES MANAGEMENT CASE STUDY

Based on the Action Shapley computation, we determine that the cut-off cardinality is equal to 5 from 15 total actions. This indicates that $P_{comp} = 99.9\%$. $\langle r_2, r_3, r_8, r_9, r_{13} \rangle$ has the highest possible mean Action Shapley value. On the other hand, $\langle r_1, r_6, r_{11}, r_{12}, r_{14} \rangle$ has the lowest possible mean Action Shapley value. As we hypothesized, the former action set produced an RL agent with a higher cumulative award of $-499$ vs $-621$. The former has a higher convergence time too: 89 vs 51. It is counter-intuitive that the former set has a higher converge time despite with a higher mean Action Shapley value. It could be due to complex system dynamics. In comparison to the baseline RL agent (cumulative reward of $-561.4$ and convergence time of 52), only the best agent performs better.

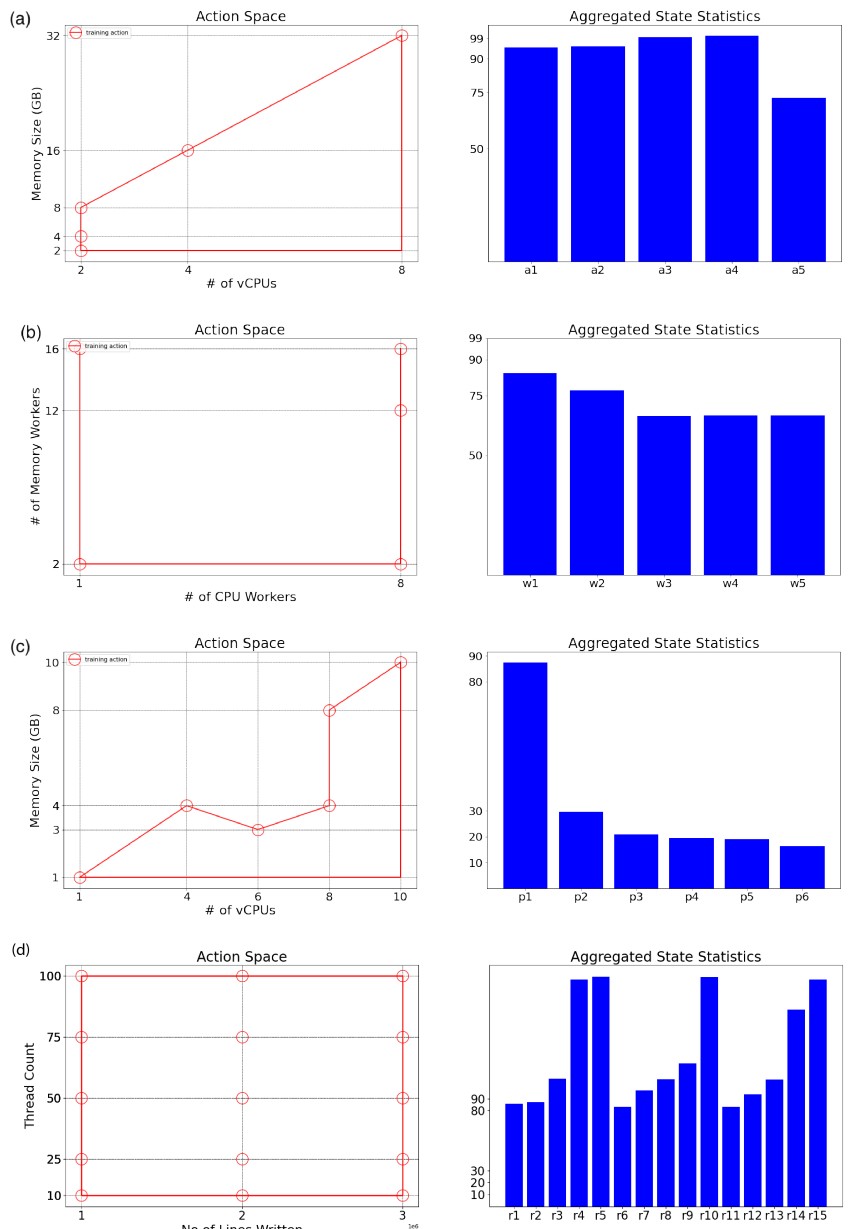

Figure 5: Different training actions and corresponding aggregated state statistics for three different case studies: (a) VM right-sizing case study. (b) Load balancing case study. (c) Database tuning case study. (d) Kubernetes management case study.

### A.7 CUT-OFF CARDINALITY

As we compute Action Shapley values for different training actions, we also compute the corresponding cut-off cardinality values. For each case study, we compute the corresponding global cardinality values as shown in Table 7. It shows for all case studies, the cut-off cardinality is lower than the number of training actions. It means the Action Shapley is more efficiency than the brute-force computation.

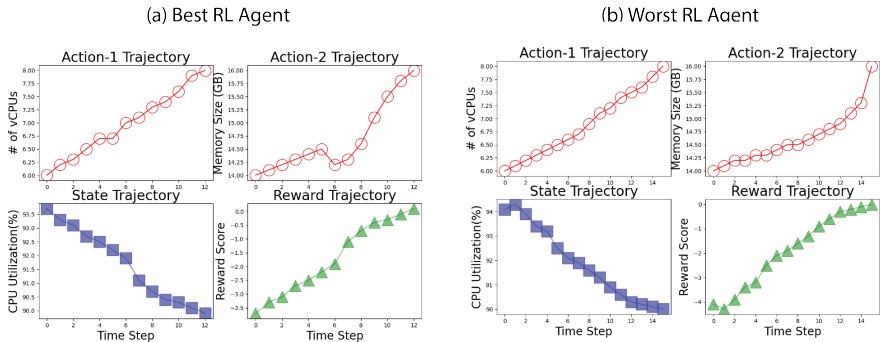

Figure 6: Action, state, and reward trajectories for the best and the worst RL agents for the VM right-sizing case study. (a) The best RL agent is produced by training actions with the highest possible Action Shapley values: $\langle a_2, a_3, a_4, a_5 \rangle$. The resulting cumulative reward is -21.9 and the convergence time is 13. (b) The worst RL agent is produced by training actions with lowest possible Action Shapley values: $\langle a_1, a_3, a_4, a_5 \rangle$. The resulting cumulative reward is -30.4 and the convergence time is 16.
.

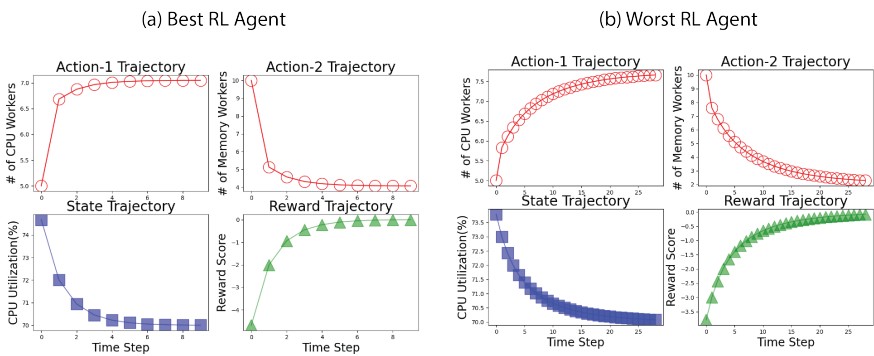

Figure 7: Action, state, and reward trajectories for the best and the worst RL agents for the load balancing case study. (a) The best RL agent is produced by training actions with the highest possible Action Shapley values: $\langle w_3, w_4, w_5 \rangle$. The resulting cumulative reward is -8.49 and the convergence time is 10. (b) The worst RL is produced by training actions with the lowest possible Action Shapley values: $\langle w_1, w_2, w_4 \rangle$. The resulting cumulative reward is -23.2 and the convergence time is 29.
.

# B  TRAINING DATA

## B.1  VM RIGHT-SIZING

## B.2  LOAD BALANCING

## B.3  DATABASE TUNING

# C  BASELINE RESULTS

The baseline study assumes the environment model for the RL agent is trained by all training actions. Table 8 shows cumulative rewards and convergence times for four case studies under the baseline condition. It shows cumulative rewards of -37.75, -9.1, -2.33, -561.4 and convergence times of 21, 12, 6, 52, for four different case studies, respectively.

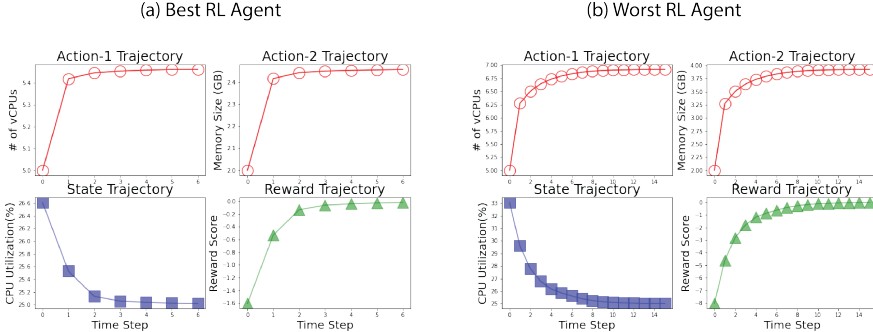

Figure 8: Action, state, and reward trajectories for the best and the worst RL agents for the database tuning case study. (a) The best RL agent is produced by training actions with the highest possible Action Shapley values: $\langle p_1, p_2, p_3, p_4 \rangle$ . The resulting cumulative reward is -2.42 and the convergence time is 7. (b) The worst RL agent is produced by training actions with the lowest possible Action Shapley values: $\langle p_1, p_2, p_5, p_6 \rangle$ . The resulting cumulative reward is -21 and the convergence time is 16.

.

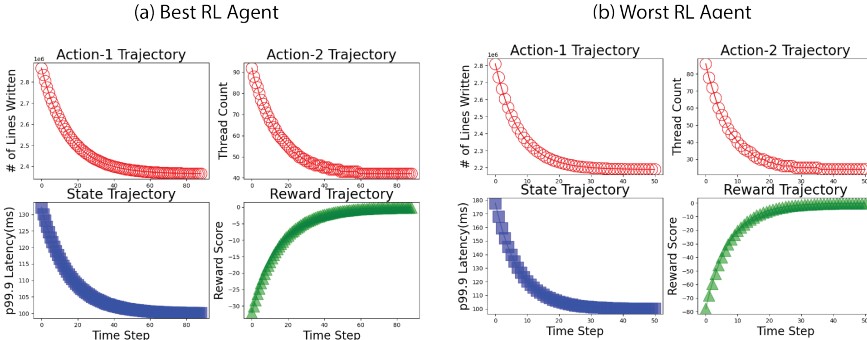

Figure 9: Action, state, and reward trajectories for the best and the worst RL agents for the Kubernetes management case study. (a) The best RL agent is produced by training actions with the highest possible Action Shapley values: $\langle r_2, r_3, r_8, r_9, r_{13} \rangle$. The resulting cumulative reward is -499 and the convergence time is 89. (b) The worst RL is produced by training actions with the lowest possible Action Shapley values: $\langle r_1, r_6, r_{11}, r_{12}, r_{14} \rangle$ . The resulting cumulative reward is -621 and the convergence time is 51.

.

Table 7: Global cut-off cardinality values and computation efficiency metrics for different case studies

| Case Study | Global Cut-off | No. of Training Actions | $P_{comp}$ |
|---|---|---|---|
| VM Right-Sizing | 4 | 5 | 50% |
| Load Balacing | 3 | 5 | 75% |
| Database Tuning | 4 | 6 | 75% |
| kubernetes Management | 5 | 15 | 99.9% |

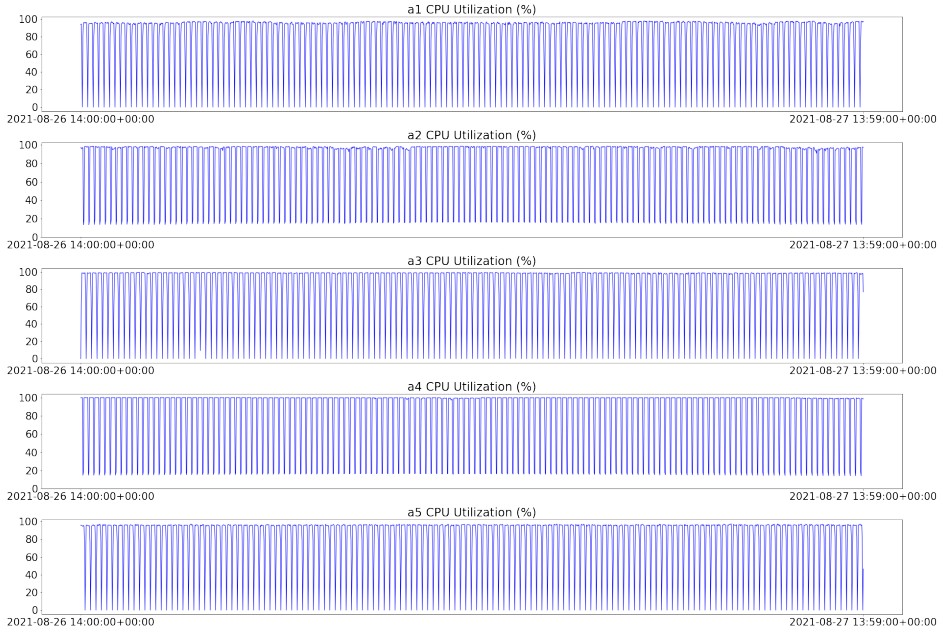

Figure 10: Training Data for VM Right-Sizing

.

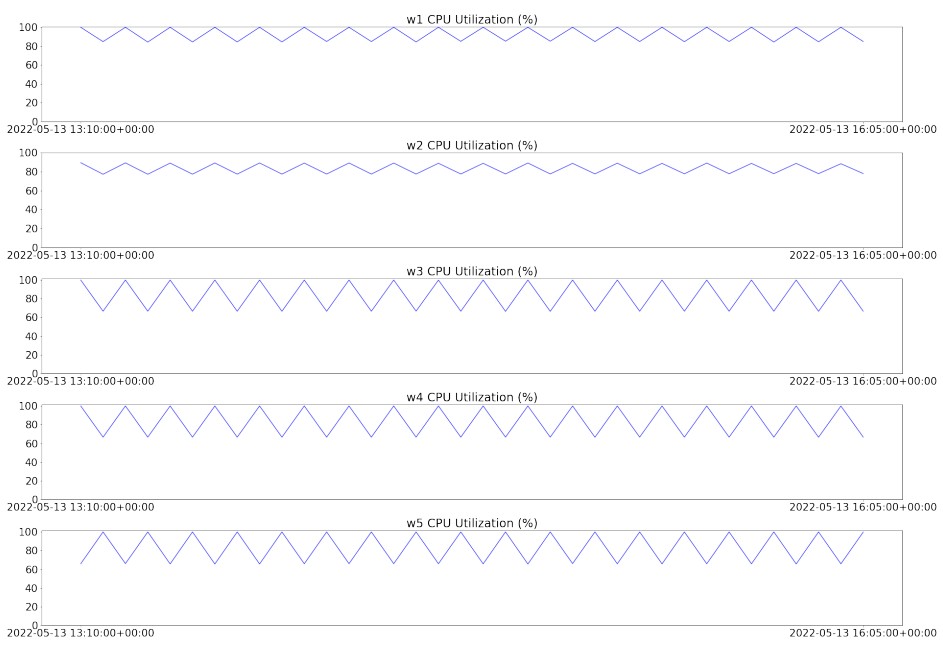

Figure 11: Training Data for Load Balancing

.

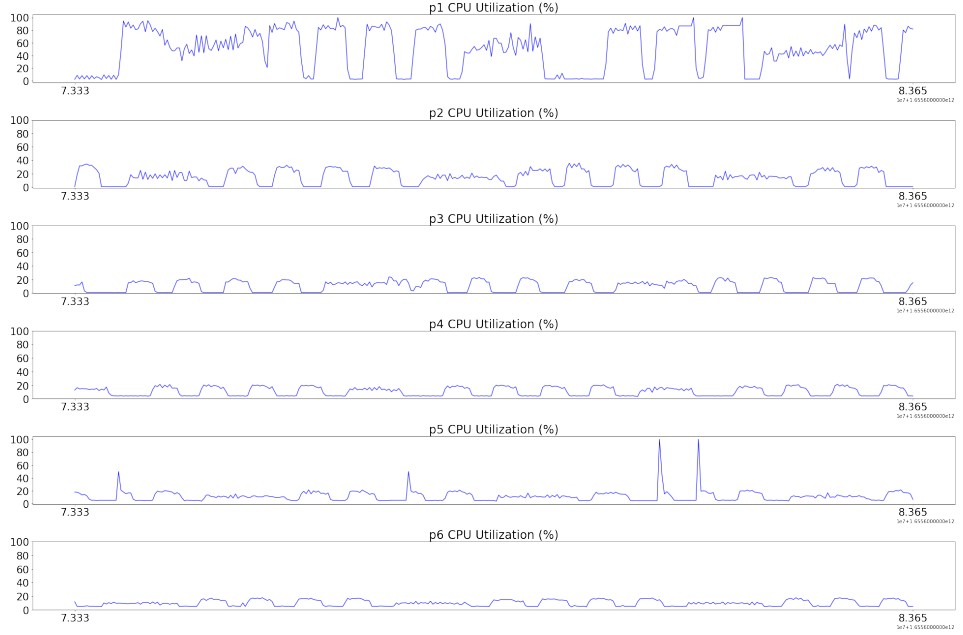

Figure 12: Training Data for Database Tuning

.

Table 8: Results from the Baseline Study

| Case Study | Cumulative Reward | Convergence Time |
|---|---|---|
| VM Right-Sizing | -37.75 | 21 |
| Load Balacing | -9.1 | 12 |
| Database Tuning | -2.33 | 6 |
| kubernetes Management | -561.4 | 52 |

