# OpenReview forum: "Action Shapley: A training data selection metric for high performance and cost efficient reinforcement learning"
_ICLR.cc/2024/Conference — Submitted to ICLR 2024_

### Official Review · Reviewer_5Ksg · 2023-10-29

**Soundness:** 2 fair
**Presentation:** 2 fair
**Contribution:** 2 fair
**Rating:** 3
**Confidence:** 4

**Summary:**

This paper studied the problem of how to select suitable training actions for reinforcement learning. The authors of this paper proposed to use the idea based on the Shapley value to guide the selection of training actions. The concept of Shapley value was proposed back in the 1950s. On several cloud computing related problems, the usefulness of selecting training actions based on the Shapley value has been experimentally demonstrated.

**Strengths:**

It is interesting and important to study possible ways of selecting useful training actions for efficient and effective reinforcement learning. The proposed use of Shapley value for selecting training actions is an interesting attempt towards solving the action selection problem.

**Weaknesses:**

The motivation for using the Shapley value for selecting training actions is not sufficiently detailed in the introduction section. To a large extent, it remains unknown why it is necessary or important to use Shapley value to guide the selection of training actions, especially when existing research works have already studied various ways of selecting training actions for model-based and model-free reinforcement learning.

There is a clear lack of review of relevant research works, especially cutting-edge technologies for selecting training actions. Hence, the real technical contribution of this paper remains highly questionable.

The mathematical definition of the action Shapley value in eq. (1) is not sufficiently clear. In fact $\phi$ is originally declared as a function of D and A, where A represents the learning algorithm. However, eq. (1) is clearly irrelevant to any learning algorithm. Meanwhile, $\mathcal{U}$ in eq. (1) is introduced as a valuation function. However, this paper did not give a clear idea regarding how $\mathcal{U}$ is defined or learned for general reinforcement learning problems. Meanwhile, eq. (3) is quite confusing. Hence, it is hard to judge on the practical value of using eq. (1) for arbitrary real-world reinforcement learning problems.

The authors stated that the best possible training action set includes as many training actions as the global cut-off cardinality with the highest Action Shapley values. However, this important algorithm design decision is only explained intuitively without clear theoretical justifications. It remains questionable whether this is the best way to select the set of training actions and what can be guaranteed by using such a set of selected actions.

For the experiments, no comparison with existing baselines was reported, making it hard to understand whether the new algorithm can achieve state-of-the-art performance on any benchmark reinforcement learning problems. The experimented benchmark problems are specific to cloud computing.  As a result, the general applicability of the proposed algorithm on various different reinforcement learning problems is doubtful.

Meanwhile, the authors stated on page 5 that they assume the next environment state depends purely on the agent action. This assumption is often wrong for reinforcement learning. Therefore, the validity of their algorithm is also questionable.

The English presentation is not sufficiently clear for many parts of this paper. Substantial changes are required to improve the presentation quality and clarity of this paper.

**Questions:**

Why is it necessary or important to use the Shapley value to guide the selection of training actions?

How is $\mathcal{U}$ is or learned for general reinforcement learning problems? Can the newly developed technique be easily applied to many different reinforcement learning problems and why?

Theoretically, why should the best possible training action set include as many training actions as the global cut-off cardinality with the highest Action Shapley values?

Why is it possible to assume that the next environment state depends purely on the agent action?

---

> ### Author Response · Authors · 2023-11-18
> **Response to Official Review of Submission525 by Reviewer 5Ksg**
>
> * The use of the Shapley value to guide the selection of training data for the environment model within a RL system is important for the following reasons: (1) Fair Attribution of Contribution: The Shapley value ensures a fair and equitable distribution of credit among the different subsets of training data. By considering all possible permutations and combinations of data points, it quantifies the marginal contribution of each data point to the model's performance. (2) Avoidance of Bias: Without a systematic method like the Shapley value, there is a risk of bias in evaluating the importance of individual data points. Some data points might be overlooked or undervalued, leading to an inaccurate representation of their impact on the model's learning. (3) Robustness and Stability: The Shapley value provides a stable and consistent measure of the contribution of each data point across different scenarios. This is crucial for maintaining robustness in model training, especially when faced with varying datasets or changes in the distribution of data. (4) Interpretability: Shapley values offer interpretability by quantifying the influence of each data point on the model's predictions. This can aid in understanding the significance of specific instances in the dataset and inform data-driven decision-making. (5) Optimal Resource Allocation: By identifying the most influential data points, the Shapley value helps in allocating computational resources more efficiently. Focusing on the most informative data can enhance the learning process and reduce unnecessary computational overhead. (6) Ethical Considerations:  In certain applications, such as sensitive domains or when dealing with user data, fairness and ethical considerations become paramount. The Shapley value provides a principled approach to treating each data point fairly, promoting ethical practices in model development.
> * In summary, employing the Shapley value for selecting training data ensures a systematic and unbiased approach to attributing importance to individual data points, contributing to a more robust, fair, and interpretable machine learning model.
> * The evaluation function, denoted as U, is delineated as the cumulative reward score corresponding to the output of the reinforcement learning (RL) agent.
> * Action Shapley exhibits a level of abstraction concerning the specific implementation details of reinforcement learning methodologies. In the context of Action Shapley, the requisite inputs consist of a set of data points and an evaluation function. This evaluation function is responsible for calculating the cumulative reward attained by the agent. Importantly, there exists no inherent impediment to the extension of Action Shapley different reinforcement learning frameworks.
> * Action Shapley exhibits a level of abstraction concerning the specific implementation details of reinforcement learning methodologies. In the context of Action Shapley, the requisite inputs consist of a set of data points and an evaluation function. This evaluation function is responsible for calculating the cumulative reward attained by the agent. Importantly, there exists no inherent impediment to the extension of Action Shapley different reinforcement learning frameworks.
> * A comprehensive exploration of the combination tree necessitates $\mathcal{O}(2^n)$ time. Consequently, there is a preference for working with a minimal set of training data points. However, it is imperative to strike a balance, as a threshold number of training data points is essential for the development of a proficient Reinforcement Learning (RL) agent. Insufficient training data points compromise the capacity to construct an effective environment model, hindering the agent's ability to fulfill its designated goal. The determination of an optimal trade-off lies in identifying the global cut-off cardinality, which represents the point of equilibrium between computational efficiency and the requisite data for robust RL agent training.
> * The stated proposition (the next environment state depends purely on the agent action) lacks general validity, and an error in the initial draft has been identified. Our empirical findings indicate that, in specific case studies such as VM right-sizing and load balancing, the subsequent state of the environment is contingent solely upon the agent's action. However, in the context of database tuning and Kubernetes management, the determination of the subsequent environment state is contingent upon both the agent's action and the prevailing state of the environment.

---

> > ### Comment · Reviewer_5Ksg · 2023-11-19
> > **Thank the authors for responding to my questions**
> >
> > Thank the authors for responding to my questions. The rebuttals however did not fully address my concerns. I think the reported research work still needs to be further improved by directly comparing with more state-of-the-art algorithms, conducting more experiments on a wide range of benchmark reinforcement learning tasks, and improving the technical clarity regarding certain equations such as eq. (1) and eq. (3).

---

> > > ### Author Response · Authors · 2023-11-20
> > > **Response**
> > >
> > > We value your feedback. Regrettably, due to time constraints, further benchmarking on additional datasets is not feasible at the moment. Nevertheless, we have incorporated PPO-PID for the purpose of comparing Action Shapley analytics with SAC-PID, the algorithm currently in use. Both PPO and SAC represent state-of-the-art in the field of reinforcement learning. Additionally, clarifications for Equation (1) and Equation (3) have been provided, and the notation section has been revised accordingly.

---

### Official Review · Reviewer_8N7h · 2023-10-29

**Soundness:** 3 good
**Presentation:** 3 good
**Contribution:** 3 good
**Rating:** 3
**Confidence:** 4

**Summary:**

The paper describes a method based on evaluation the Shapley value of
the actions, in order to rank them and select high value actions.

Evaluation the Shapley value of an action requires summing over all
possible subsets of value functions with and without that action. This
is clearly computationally very expensive so the authors propose an
incremental approach where subsets are tested for failures under a given
\epsilon parameter. If all, but \epsilon subsets for a given
cardinality produce unsuccessful RL agents, the computation is
terminated. In this sense, they can cut-off some evaluations.

The proposed approach is tested of four similar domains with a
relative small set of actions. The authors showed that given the
Shapley values for the actions, in general, the system can achieve
better and faster performance.

**Strengths:**

- Apply Shapley values to the selection of actions
- Show that some actions may not be needed, producing saving is training
time, without affecting performance.
- Selecting the actions with best Shapley values have in general
better performance

**Weaknesses:**

- Computationally expensive
- Applicable to very simple domains (discrete and deterministic with
few actions)

**Questions:**

It is not clear why the authors mention that trial-and-error for RL
in domains like Go or StarCraft are relatively inexpensive.

All the tests are performed in very similar domains, which questions
the applicability to other domains.

It is not clear how to select \epsilon. A high \epsilon means more
computation, while a low \epsilon.

It seems to be applicable only to discrete domains with a small number
of possible actions. Also, the tests are performed on deterministic
environments. Such conditions seem very restrictive for real world or
even simple, domains.

Evaluating the Shapley values is still computationally expensive, even
with the proposed algorithm. It is not clear how much an agent gains
with this approach. Once the values are known, the gain is quite
clear, however, the authors do not report how expensive is to obtain
such values.

The use of the Shapley value is not new in the literature. The main
difference in this paper is to use it for the selection of actions.

Some terms are not properly described in the paper, e.g., g(a_t), \phi
The paper has several English errors that need to be corrected.

---

> ### Author Response · Authors · 2023-11-18
> **Response to Official Review of Submission525 by Reviewer 8N7h**
>
> * Trial-and-error in reinforcement learning (RL) for domains like Go or StarCraft is relatively inexpensive for several reasons: (1) Simulated Environments: In games like Go or StarCraft, it is often possible to create highly realistic simulations of the environment. Simulations allow agents to explore and learn without interacting with the real world, reducing the cost and risks associated with physical experimentation.
> (2) Fast Iteration: Simulated environments enable fast and efficient iteration. Agents can undergo numerous trials in a short amount of time, accelerating the learning process. This rapid iteration is particularly advantageous for RL algorithms, allowing them to explore a large state space and discover effective strategies quickly. (3) Parallelization: Many RL algorithms can be parallelized, enabling multiple agents to explore different possibilities simultaneously.  (4) Low Resource Costs: Simulations typically require fewer resources than physical experiments. Running simulations is computationally less expensive than deploying real-world agents, making it more cost-effective to perform extensive trial-and-error in simulated environments. (5) Reproducibility: Simulated environments provide a high level of reproducibility. Researchers and practitioners can precisely replicate experiments, facilitating the comparison of different algorithms and strategies.  (6) Safety: In domains like StarCraft, trial-and-error in a simulated environment avoids the potential risks associated with deploying agents in the real world. RL agents can learn complex strategies and tactics without the concern for safety issues or unintended consequences.
>
> * The conducted tests are executed within the cloud computing infrastructure; however, it is noteworthy that no explicit reliance on specific system properties is made. The dataset is intentionally randomized across diverse systems, encompassing Virtual Machines (VMs) as computing systems, load balancers as networking systems, databases as storage systems, and Kubernetes as a container orchestration system. Each of these systems exhibits distinct time constants and seasonality patterns, as elucidated in the Appendix. For instance, the training data for Database Tuning (Figure 12) manifests random variability. In contrast, the training data for both VM right-sizing (Figure 10) and load balancing (Figure 11) adopt a triangular distribution with disparate temporal characteristics.
>
> * $\epsilon$ is a user-defined parameter. A heightened value of  $\epsilon$ signifies a proclivity toward exploring a greater array of training data subsets and engaging in a more aggressive pursuit of an optimal training subset. Conversely, a diminished $\epsilon$ indicates a preference for the exploration of fewer training data subsets, thereby facilitating a more expeditious convergence. This dynamic reflects a discernible exploration-exploitation trade-off.
> * The domain under consideration exhibits a continuous-state space paired with a discrete-action space. The diversity of potential action values is evident in the trajectories depicted in Figures 6 through 9 in the Appendix. The training set utilized for the environment model comprises 4, 5, 6, and 15 training actions, each accompanied by a corresponding state time series, encompassing 1440 data points. This configuration closely mirrors the intricacies of real-world systems.
>
> * We have incorporated a discussion on computational complexity on Page 4, with specific reference to the discourse surrounding Equation 2. The computational complexity of Algorithm 1 is situated between $\mathcal{O}(2^n)$, denoting the worst-case scenario, and $\mathcal{O}(\epsilon)$, reflecting the best-case scenario. Combining these two extremes of $\mathcal{O}(2^n)$ and $\mathcal{O}(\epsilon)$, the performance of Algorithm 1  can be represented as a ratio of the exponential of global cut-off cardinality and the exponential of the total number of training data points subtracted from $1$, as shown in Equation 2.
>
> * Our distinctive contribution involves the innovative application of the Shapley value in selecting training data for the environment model training of a Reinforcement Learning (RL) system. The key challenges stem from: (1) Combinatorial Explosion: RL environment often deals with large state and action spaces, leading to a combinatorial explosion of possible coalitions (combinations of agents) for which Shapley values need to be calculated. (2) Sequential Decision Making: RL involves sequential decision making, where the impact of an agent's action may not be immediately apparent but could unfold over time. This temporal aspect introduces complexities. (3) Model Uncertainty: RL often involves uncertainty, both in terms of the environment dynamics and the learned model.
> * The manuscript underwent comprehensive proofreading procedures, and an additional section dedicated to Notation has been incorporated within the Appendix.

---

> > ### Comment · Reviewer_8N7h · 2023-11-22
> > **Thanks for your reply, there are still some concerns**
> >
> > I would like to thank the authors for their replies to my concerns
> > about this paper. While some of them have been satisfactorily
> > addressed, I believe that some others remain valid. I still believe
> > that the tests are performed on very similar domains, with slight
> > variations, and that the approach is only applicable to discrete action
> > spaces and deterministic environments, which limits its
> > applicability. Although the authors showed the cutoff benefits in the
> > action selection of the proposed approach, these can be more
> > difficult to attain as the domain becomes more complex.
> > It is not clear how general or what are the main limitations of the
> > proposed approach.

---

> > > ### Author Response · Authors · 2023-11-22
> > > **Response**
> > >
> > > We express gratitude for the provided feedback and would like to present subsequent inquiries and observations:
> > >
> > > 1. In the event of introducing stochasticity to the test environments, do you posit that the overall quality of the study would have been enhanced?
> > >
> > > 2. In accordance with my comprehension, a continuous action space pertains to the capability of a Reinforcement Learning (RL) agent to select an action from a continuum of possible values. It is acknowledged that within cloud systems, actions predominantly manifest as discrete entities. Future iterations of our work will encompass additional investigations in this domain.
> > >
> > > 3. Could you kindly furnish an illustrative example of a complex domain? This would enable us to refine our manuscript to better align with the intricacies of such domains.

---

### Official Review · Reviewer_mr6C · 2023-10-30

**Soundness:** 2 fair
**Presentation:** 1 poor
**Contribution:** 1 poor
**Rating:** 3
**Confidence:** 5

**Summary:**

This paper aims to give the understanding of 'Superior interpretability demands granular under- standing of the differential impact of the training actions on the resulting RL agent performance'. To achieve it, this work provides an agnostic metric for the selection of training actions and provides a feasible method to calculate. The authors demonstrate the effective of their method in real-world tasks.

**Strengths:**

1. This paper targets a valuable problem, that is selecting high performance training action set for RL.
2. This work conducts experiments on real-world tasks to verify their effectiveness.

**Weaknesses:**

1. The presentation of this work is very poor. I have the following suggestions to greatly improve readability: (1) Add a Background and Notation section before methodology section. It is difficult to understand this method directly without relevant background knowledge. (2) The text description of the article is divided into appropriate paragraphs. This manuscript has only one paragraph for almost every chapter, which makes it tiring for the reader. (3) Table 3, Table 4, and Table 5 must be carefully arranged. (4) A label should be added to the display of pictures.

2. The motivation of this paper is not explained well. I cannot get the deep insight from the current version of this manuscript. I believe this manuscript was hastily completed, and I believe the author may have solved some interesting problems. Please revise this manuscript according to my comments above before reviewing it.

**Questions:**

Please refer to the above weakness.

--------

Thanks for the explanation. I maintain my score.

---

> ### Author Response · Authors · 2023-11-18
> **Response to Official Review of Submission525 by Reviewer mr6C**
>
> * The authors express sincere gratitude for the valuable feedback received. In response to the limitations imposed by page constraints, a new subsection titled "Background and Notation" has been incorporated at the outset of the Methodology section. Furthermore, to effectively navigate the challenges posed by page limitations, a dedicated Notation section has been appended to the document's Appendix.
> * We appreciate your observation regarding the paragraph division and understand your concern about the potential strain it may place on the reader. In response to your suggestion, we have revised the manuscript to enhance readability by incorporating more nuanced paragraphing, ensuring a smoother flow between ideas. We trust that these adjustments contribute positively to the overall reading experience with the specified constraints of page limit.
> * The tables are appropriately arranged.
> * All figures have labels.
> * The provided feedback has been valued, leading to a comprehensive revision of the manuscript. Our objective is to establish a principled selection metric for training data pertaining to the environment model within a reinforcement learning system. Action Shapley is distinguished among alternative selection methodologies such as LIME due to its advantages in sample efficiency, model agnosticity, interpretability, fairness, and stability.
> * While the use of the Shapley value has been documented in extant literature, our contribution lies in its novel application to the domain of training data selection for the environment model of a reinforcement learning (RL) system. The integration of Shapley value in this context presents several challenges: (1) Combinatorial Explosion: RL often deals with large state and action spaces, leading to a combinatorial explosion of possible coalitions (combinations of agents) for which Shapley values need to be calculated. This computational complexity can be prohibitive for exact Shapley value computation, especially in complex environments. (2) Sample Inefficiency: Estimating Shapley values requires evaluating the marginal contribution of each agent across all possible coalitions. In RL, obtaining accurate estimates of these marginal contributions may require a large number of samples, leading to sample inefficiency, which can be a significant drawback in terms of computational resources. (3) Sequential Decision Making: RL involves sequential decision making, where the impact of an agent's action may not be immediately apparent but could unfold over time. This temporal aspect introduces complexities in attributing contributions to individual agents, as their actions may interact in non-trivial ways. (4) Model Uncertainty: RL often involves uncertainty, both in terms of the environment dynamics and the learned model. Shapley values rely on a well-defined cooperative game, and uncertainties can introduce challenges in accurately modeling and attributing contributions. (5) Computational Resources: Calculating Shapley values can be computationally expensive, and for real-time applications or resource-constrained environments, this may pose a limitation.
> * We handle this problem using a randomized dynamic algorithm, acknowledging the trade-offs between accuracy and computational feasibility.
> * The manuscript underwent comprehensive proofreading procedures, and an additional section dedicated to Notation has been incorporated within the Appendix. The authors express gratitude for the thorough review conducted on the revised manuscript.

---

### Official Review · Reviewer_oVx8 · 2023-11-06

**Soundness:** 2 fair
**Presentation:** 2 fair
**Contribution:** 2 fair
**Rating:** 3
**Confidence:** 5

**Summary:**

Authors present a Shapley value calculation framework integrable in arbitrary RL algorithms. The randomized variant of the Shapley value computation method is applied in four settings.

**Strengths:**

- Originality:
To the best of my knowledge, this is the first work to directly apply Shapley value constructs on action and use the computed values as a selection criterion.

- Quality:
Some interesting real-world application scenarios have been set up for evaluation.

- Clarity:
Evaluation setting is straightforward, and the randomized Shapley value calculation is explained clearly, with few-action examples.

- Significance:
It is difficult to measure the significance of the paper, as the comparative analyses are weak.

**Weaknesses:**

Comparative analyses across several dimensions are rather lacking.
Along the conceptual axis, state-action values (Q-values) have long served as action selection criterion, but there is no mention as to how the Shapley construct offers any theoretical advantages or empirically observed performance gain. Moreover, in the full RL setting, the marginal contribution of any action is assumed to be under the influence of the state, hence q-values are a mapping from a state-action pair to a real value. However, the authors recede (and actually collapse) the problem into a contextual multi-armed bandit, where “the next environment state is determined purely by the agent action”. One natural baseline in the CMAB setting would be the UCB algorithm and its variants, but none is compared against.
Along the evaluation axis, while the provided examples are motivating, some of the better known scenarios could help position the work more strongly.
Along the algorithmic analysis axis, it is hard to exactly measure the effects of the randomization, as there are only very few actions to begin with. Perhaps some asymptotic analysis between the baseline and the proposed algorithm could build a stronger scalability argument.

**Questions:**

What is a training action? How is it different from an action? What is the role of this new terminology?
What is meant by “model-based” in the Conclusion section? Do we start with the MDP fully known? If that’s the case, then what is the advantage of action Shapley over dynamic programming methods?
If we have so few actions to begin with, what is the advantage of action Shapley over Monte Carlo tree search, which will provide an exact solution?
How are the action Shapley values aggregated over different states?
How does action Shapley fare in, say, DQN?

---

> ### Author Response · Authors · 2023-11-18
> **Response to Official Review of Submission525 by Reviewer oVx8**
>
> * A training action is a segment of a training data point that signifies the action associated with the compilation of that specific data point. In the context of this paper, each training data point is conceptualized as an ordered pair consisting of an action value and the corresponding state time series, all pertaining to a specified initial condition. This formulation is expressed as follows: $\langle a, T  | a_{o}, T_{o} \rangle$.
>
> * “Model-based” means using a model-based reinforcement learning system. Model-based reinforcement learning (MBRL) is an approach in the field of reinforcement learning (RL) where the agent builds an internal model of the environment to make decisions. In traditional RL, agents directly interact with the environment, learn from trial and error, and update their policies based on the observed rewards. In contrast, MBRL involves the creation and utilization of a model that approximates the dynamics of the environment.
>
> * No, we don’t start MDP fully known. That is why we build an environment model.
>
> * While Monte Carlo tree search (MCTS) can provide an exact solution with a sufficient number of simulations, Shapley values offer distinct advantages, particularly in scenarios with limited data points. (1) Data Efficiency: Shapley values require fewer samples compared to Monte Carlo tree search for certain applications. This becomes crucial when more sample collection may not be feasible or practical. (2) Interpretability: Shapley values provide a clear and intuitive way to understand the contribution of each action or player in a cooperative game. This interpretability is valuable for decision-makers who need to comprehend the impact of individual actions. (3) Fairness and Stability: Shapley values satisfy desirable properties such as fairness and stability in cooperative games. They distribute the value of a coalition among its members in a way that is considered fair, and they are not influenced by the order in which contributions are made. This stability is beneficial in situations where consistency is important. (4) Model-Agnostic: Shapley values are model-agnostic, meaning they can be applied to various types of models and systems. This flexibility is advantageous when working with different RL environments or models.
> (5) Applicability to Cooperative Settings: Shapley values are particularly well-suited for cooperative game settings where the impact of each agent's actions contributes to a shared outcome. This is different from MCTS, which is often applied in competitive or adversarial settings.
> While MCTS provides an exact solution with a sufficient number of simulations, Shapley values may offer practical benefits in terms of efficiency, interpretability, fairness, and applicability to cooperative scenarios, especially when dealing with limited data points.
>
> * Action Shapley is not a metric designed for time series points. The Action Shapley value is computed for a training data point during the training process of an environment model for a Reinforcement Learning (RL) system. The assessment entails the calculation of cumulative rewards within a reinforcement learning loop, which terminates when an aggregate state statistic exceeds a predefined threshold. Different state aggregation methods are utilized in various case studies. For instance, p50 is employed for VM right-sizing, p5 for load balancing, p50 for database tuning, and p99.9 for Kubernetes management. Page 2 of the manuscript discusses how Action Shapley is computed in reference to Equation 1.
>
> *  Action Shapley serves as an impartial selection metric for the training data points used in the environment model of a Reinforcement Learning (RL) system. Deep Q Networks (DQN) is a category of RL algorithms specifically designed to address challenges where an agent must make sequential decisions over time to maximize cumulative rewards. Within the framework of Action Shapley, the essential inputs comprise a set of data points and an evaluation function responsible for computing the cumulative reward achieved by the agent. Notably, there is no inherent limitation to extending the application of Action Shapley to Deep Q Networks (DQN) or analogous reinforcement learning frameworks.
>
> *  The  Q-value facilitates the agent in evaluating the desirability of diverse actions across varying states, thereby guiding its decision-making process to achieve an optimal overall outcome over time. Consequently, the Q-value holds relevance within the reinforcement learning loop, while Action Shapley emerges as an equitable metric for the selection of training data for the environment model in a reinforcement learning system.
>
> * The updated manuscript includes a dedicated section addressing the analysis of algorithmic complexity in page 3. The computational complexity of Algorithm  is situated between $\mathcal{O}(2^n)$, denoting the worst-case scenario, and $\mathcal{O}(\epsilon)$, reflecting the best-case scenario.

---

> > ### Comment · Reviewer_oVx8 · 2023-11-22
> > **Thank you**
> >
> > I have read the authors' response.
> >
> > The following questions remain unaddressed, and I keep my score:
> >
> > 1. The purpose of defining a new terminology, "training action" has not been described. In fact, the very attempt to explain training action introduces yet more terms, "segment" and "training data point", neither of which is defined. Their difference from well-defined and well-established terms "trajectory" and "experience sample (s, a, r, s')" is also not described.
> >
> > 2. If the authors are doing model-based RL as they claim, then there is no longer a need to "create and utilize a model that approximates the dynamics of the environment". Many RL works, including the seminal work cited in the paper by Sutton and Barto, views MBRL as a class of RL algorithms operating on a given model, not an approximated one.
> >
> > 3. Again, MBRL assumes MDP known, by definition. The authors' description of their methodology do not add up. If the presented method is indeed an instance of MBRL, then there is no need to build a model separately. According to the authors' response, a model is built nevertheless, but the advantages of building that model are not described. That is, why build a model when MDP is given?
> >
> > 4. Advantages of using Action Shapley over MCTS are not described. Authors' response instead describes Shapley values in general (not action Shapley). Whatever advantages Shapley values bring are not a contribution by the submission; they are contributions by the original designers of the very concept of Shapley values.
> >
> > 5. How exactly there is one value associated to one action is not described. The "goodness" of an action naturally varies with the state, but the aggregation along the states axis remains largely unaddressed.
> >
> > 6. How Action Shapley fares in DQNs is not described. Only the potential applicability of Action Shapley (which was not a part of my question) is reiterated.
> >
> > 7. How exactly Action Shapley construct would be advantages over Q-values is not described. Only a brief description of what Q-values are (which was not a part of my question) is reiterated.

---

> ### Author Response · Authors · 2023-11-22
> **Response**
>
> Thanks for the feedback. A few follow-on comments:
>
> 1. The training action denotes the action component within a training data point. The manuscript defines a training data point as follows: "Consequently, a training data point is appropriately represented as: $\mathcal{D} = \{(s^n_t; (s^n_{t-1}, a^n_{t-1}))\}.$" Herein, $a^n_{t-1}$ of $\mathcal{D}$ is specifically identified as the training data point.
> 2. In the model-based reinforcement learning (MBRL), the environment model is not inherently accessible in all cases. The principal objective of this study is to contribute to the refinement of the environment model by facilitating equitable selection of training data.
> 3. The model is built to simulate the environment. We have assumed MDP. That is the basis of the training data formulation. Here is the excerpt from page 2 of the paper: 'In the realm of Reinforcement Learning (RL), the primary objective of a goal-seeking agent lies in the maximization of a cumulative reward score through dynamic interactions with its environment. These interactions are succinctly encapsulated in a temporal sequence denoted as $\langle state, action, reward \rangle$ trajectories, expressed as $ \langle s_{0}, a_{0}, r_{1}, s_{1}, a_{1}, r_{2}, s_{2}, a_{2}, r_{3} ...\rangle $. Under the assumption of a Markov decision process, the conditional joint probability distribution of reward ($r_t$) and state ($s_t$) is contingent solely upon the antecedent state and action, denoted as $p(s_{t}, r_{t} | s_{t-1}, a_{t-1})$. In the context of model-based RL systems, the underlying supervised model strives to capture this distribution. This necessitates a training corpus comprising two pairs: firstly, $(s_{t-1}, a_{t-1})$, followed by $(s_{t}, r_{t})$. In the majority of cases, the reward score can be derived from the state: ${s_{t} \mapsto r_{t}}$. Consequently, a training data point is aptly represented as:  $ \mathcal{D} = \{(s^n_t;  (s^n_{t-1}, a^n_{t-1}))\}_{n}.$"
> 3. (a). There appears to be a conflation between the conceptualization of the environment model and the underlying reinforcement learning (RL) model. Specifically, the environment model functions as a simulation, providing a replicated representation of the external system with which the RL agent interacts.
> 4. Action Shapley constitutes a training data selection metric inspired by the Shapley value. It specifically tailored for addressing the intricacies associated with the problem of the environment modeling in the context of reinforcement learning. The environment serves as an encompassing framework encapsulating the external system with which the agent engages, thereby exerting a pivotal influence on the configuration of the learning process. Given its adherence to the Shapley value framework, Action Shapley inherits analogous advantages to the foundational Shapley value with respect to MCTS.
> 5. Various state aggregation methods have been employed across distinct case studies. Specifically, the p50 method is applied in the context of virtual machine (VM) right-sizing, while the p5 method is utilized for load balancing. Additionally, the p50 method is employed in the domain of database tuning, and the p99.9 method finds application in Kubernetes management. A comprehensive exposition of these methodologies is presented in Section 3: Data Collection and Implementation Section.
> 6. The Deep Q-Network (DQN) constitutes a representative instance of a reinforcement learning (RL) algorithm, while Action Shapley stands as a training data selection metric employed for the purpose of training data selection within the context of an environmental model. It is noteworthy that there exists no inherent impedance to the integration of Action Shapley within the framework of DQN.
> 7.  The Q-value functions as a quantitative measure for evaluating the decision-making policy of a Reinforcement Learning (RL) agent when selecting an action (A) to execute within a given state (S). In contrast, Action Shapley stands as a training data selection metric for the environmental model. It is essential to emphasize that these conceptual constructs are fundamentally disparate in their scopes and consequential implications.

---

### Meta-Review · Area_Chair_cUQY · 2023-11-28

**Metareview:**

**Summary**: This paper proposes an RL method that selections actions by computing the Shapley value (summing over all possible subsets of value functions with and without that action). Experiments demonstrate the method on cloud computing applications.

**Strengths**: The reviewers appreciated that the method was applied to real-world problems. They also noted that the method is novel and seems to outperform some prior methods.

**Weaknesses**:
Most reviewers felt that the paper needed revisions to improve clarity. Multiple reviewers suggested to clarify the motivation for the paper. The reviewers also felt that comparing to additional baselines and on more complex tasks would strengthen the paper. There was some general confusion about whether the method assumes a known model of the environment, though this was clarified by the authors during the discussion.

Overall, it seems like the reviewers did appreciate the novelty of the paper. After revisions and adding a few new experiments and more complex tasks, this could make a strong submission to a future conference.

**Justification For Why Not Higher Score:**

Many of the reviewers had concerns about clarity and limited experiments; none voted for accepting the paper.

**Justification For Why Not Lower Score:**

N/A

---

### Decision · Program_Chairs · 2024-01-16

Reject